# Alterations in Hepatocellular Carcinoma-Specific Immune Responses Following Hepatitis C Virus Elimination by Direct-Acting Antivirals

**DOI:** 10.3390/ijms231911623

**Published:** 2022-10-01

**Authors:** Shihui Li, Eishiro Mizukoshi, Kazunori Kawaguchi, Miyabi Miura, Michiko Nishino, Tetsuro Shimakami, Kuniaki Arai, Taro Yamashita, Yoshio Sakai, Tatsuya Yamashita, Masao Honda, Shuichi Kaneko

**Affiliations:** Department of Gastroenterology, Graduate School of Medicine, Kanazawa University, Kanazawa 920-8641, Japan

**Keywords:** epitope, immunotherapy, F protein, PD-1, cancer

## Abstract

Direct-acting antivirals (DAAs) have recently revolutionized the eradication of chronic hepatitis C virus (HCV) infection. However, the effects of DAAs on the development of hepatocellular carcinoma (HCC) remain unknown. Therefore, the present study aimed to investigate immune responses to HCC influenced by DAAs in HCV-infected patients and elucidate the underlying mechanisms. We compared immune responses to 19 different HCC-related tumor-associated antigen (TAA)-derived peptides and host immune cell profiles before and 24 weeks after a treatment with DAAs in 47 HLA-A24-positive patients. The relationships between the different immune responses and phenotypic changes in immune cells were also examined. The treatment with DAAs induced four types of immune responses to TAAs and markedly altered host immune cell profiles. Prominently, reductions in the frequencies of PD-1+CD4+ and PD-1+CD8+ T cells by DAAs were associated with enhanced immune responses to TAAs. The HCV F protein was identified as contributing to the increased frequency of PD-1+ T cells, which may be decreased after eradication by DAAs. DAAs altered the immune responses of patients to HCC by decreasing the frequency of PD-1-expressing CD4+ and CD8+ T cells.

## 1. Introduction

Hepatocellular carcinoma (HCC) is an aggressive hepatic malignancy with poor prognosis. The chronic progression of hepatitis C virus (HCV) infection, which can develop into chronic liver disease such as hepatitis and cirrhosis, has been identified as the high risk for occurrence of HCC [1,2,3]. Some studies even point out the possibility of its direct carcinogenic effects [1].

The combination of two or three direct-acting antivirals (DAAs), which are very effective for viral eradication, has been shown to cure more than 90% of HCV-infected patients, as well as patients with no response to prior treatment, such as interferon (IFN)-based therapy [4,5]. Previous reports suggest that DAAs treatment can induce broad phenotypic change in immune T cells such as the expression of molecular PD-1 and CTLA-4 [6,7]. In recent years, the concept of immunological scarring, which means chronic infection with HCV might leave lasting effects on the immune system, is being acknowledged [7,8]. These compromised immune mechanisms must interact with the chronic inflammation of persistent infection, thus susceptibility to malignancy.

In cases where HCV RNA was successfully eliminated by interferon therapy, hepatitis subsided and the development of hepatic fibrosis and hepatocarcinogenesis were suppressed [9]. Although there are some negative reports on whether HCV elimination by DAAs therapy is as effective as IFN therapy in suppressing hepatocarcinogenesis [10,11,12], there are increasing reports that it is as effective as IFN therapy in suppressing hepatocarcinogenesis [13,14,15]. Therefore, the DAAs might influence and even amplify HCV-induced immune changes and affect anti-tumor immunity.

However, the detailed mechanism by which viral elimination achieved by DAAs treatment reduces the risk of developing HCC remains unclear. Furthermore, it has been reported that changes in cell surface markers of T cells occur after viral elimination [7,16], but it is not clear how the changes in host immune cells caused by removal of chronic viral antigen stimulation affect the immune response to HCC.

In the present study, to elucidate the influence of DAAs for the immune response to HCC in HCV-infected patients and the mechanism underlying it, we performed a comparative analysis from two aspects: immune responses to HCC-related TAA-derived peptides and host immune cell profiles before and 24 weeks after treatment.

## 2. Results

### 2.1. Patient Profile

To investigate the influence of the treatment with DAAs on immune responses to HCC, the PBMCs of 47 chronic HCV-infected patients who received ASV and DCV were collected before and 24 weeks after treatment. The baseline characteristics of these patients before treatment are listed in Table 1. Of the 47 patients in this study, 6 had a history of HCC prior to treatment with DAAs.

All patients were divided into two groups based on the therapeutic outcomes of DAAs: SVR (N = 44), and non-SVR (N = 3). No significant difference was observed in clinical features between the two groups, except for the NS5A Y93H mutation. Before treatment, the NS5A-Y93 mutation was detected in none of the patients in the SVR group, but in two out of the three patients in the non-SVR group (*p* = 0.003).

### 2.2. The Treatment with DAAs Induced Different Immune Responses to TAA-Derived Peptides

To clarify the impact of the treatment with DAAs on immune responses to HCC, IFN-γ production by HCC-related TAA-specific T cells was measured in all patients before and after the treatment using the ELISPOT assay. A total of 19 peptides derived from 14 different HCC-related TAAs and 1 peptide derived from CMV pp65 as a control were used to examine the immune responses of patients (Figure 1A and Appendix A). Numbers of patients with a positive T cell response to each peptide were observed in 0 to 12 out of 47 (0.0–25.5%) before treatment and in 1 to 16 out of 47 (2.1–34.0%) after treatment (Figure 1B). The number of patients with a positive response to each peptide increased after treatment, except for the peptides MRP3_692_, hTERT_461_, NY-ESO-1_158_, and IMP-3_508_. Positive responses were not observed before treatment, but after treatment, 3 and 2 patients showed a positive response to the peptides GPC3_298_ and SCCA_112_, respectively. The number of patients with a positive response to each peptide after DAAs treatment was significantly higher than that before (*p* < 0.01).

To elucidate the immune responses to peptides influenced by DAAs, we differentiated the immune responses of patients by individual comparisons of specific spots, which reflected the number of activated T cells, before and after the treatment with DAAs (Appendix A). A significant increase was defined as the number of spots after treatment being ≥10 and two-fold higher than that before treatment. In contrast, a significant decrease in the number of spots was defined as a pre-treatment spot count of 10 or more, and a post-treatment spot count of less than one-half of the pre-treatment spot count (Figure 1C). The results of the comparison showed that the treatment with DAAs induced different immune responses to TAA-derived peptides, based on which 47 patients were categorized into four groups: increased group (n = 11), mixed group (n = 18), decreased group (n = 9), and unchanged group (n = 9) (Figure 1D). Among the patients analyzed, 29/47 (61.7%) patients, which were the total of patients belonging to increased and mixed groups, showed an increase in the number of IFN-γ-producing T cells for at least one TAA-derived peptides after the treatment. On the other hand, the frequency of T cells specific for the control peptide increased in only 4/47 (8.5%) patients (Appendix A).

### 2.3. The Treatment with DAAs Significantly Altered Immune Cell Profiles

We previously reported that the frequency of TAA-derived peptides was associated with the host immune cell profiles [17,18]. Therefore, we performed a comparative analysis of T cell and myeloid-derived suppressor cell (MDSC) profiles before and after the treatment with DAAs using flow cytometry by assessing the expression of 20 different molecules that potentially affect HCC-specific host T cell responses (Figure 2).

CD4+ T cells were divided into different subpopulations by CD45RA and Foxp3 for a more detailed analysis: naïve CD4+ T cells (CD4+CD45RA+), naïve non-Treg CD4+ T cells (CD4+CD45RA+Foxp3-), naïve Treg (CD4+CD45RA+Foxp3+), memory CD4+ T cells (CD4+CD45RA-), memory non-Treg CD4+ T cells (CD4+CD45RA-Foxp3-), and effector Treg (CD4+CD45RA-Foxp3++) [17]. CD8+ T cells were divided into two subpopulations based on CD45RA: naïve CD8+ T cells (CD8+CD45RA+) and memory CD8+ T cells (CD8+CD45RA-) (Figure 2A). According to these differentiations, the expression of PD-1, CTLA-4, CD25, CCR6, CXCR3, CCR4, 4-1BB, OX40, and CD80 was separately analyzed on the surface of each subpopulation (Figure 2B). Two major subpopulations of MDSCs were simultaneously identified by the expression of CD14, CD15, CD33, CD11b, and HLA-DR (Figure 2C). CD14+HLA-DR- MDSCs, labeled as M-MDSC and G-MDSC, were defined as CD14-, CD15+, CD33+, and CD11b+ cells. PD-L1 expression levels were then assessed in each subpopulation.

Comprehensive comparisons revealed significant differences in host immune cell profiles before and after the treatment with DAAs (Figure 3). In all patients, the frequencies of T cells expressing the tested molecules after treatment mostly showed a significant decrease (Figure 3A). Consistent with previous findings [6], DAAs treatment induced broad phenotypic changes.

### 2.4. Relationship between Immune Cell Profiles and HCV Eradication

Next, we investigated whether the removal of HCV by DAAs resulted in phenotypic changes in T cells and MDSCs. A comparative analysis of changes in T cell and MDSC profiles in the SVR (n = 44) and non-SVR (n = 3) groups was performed (Figure 3B,C and Appendix A). In the analysis of the SVR group, the frequencies of T cells and MDSCs expressing each immune-related molecule varied over a wide range of species, which was similar to the results of the overall analysis. On the other hand, in the analysis of the non-SVR group, changes in T cells and MDSCs were limited to some molecules. These results provided further support for immune cell profiles being altered by the eradication of HCV after the treatment with DAAs. Among the nine tested T cell surface molecules in the present study, the frequency of PD-1-expressing T cells decreased after viral clearance in most phenotypes, including CD4, memory CD4, and CD8. In contrast, in the non-SVR group, no significant decrease in the frequency of these cells expressing PD-1 was noted, while an increase of naïve non-Treg CD4+ T cells expressing PD-1 was observed. These results suggest that HCV eradication by DAAs plays a vital role in inhibiting the PD-1/PD-L1 pathway. The remarkably decreased frequency of PD-1+ T cells, which play a fundamental role in T cell activation against tumors, suggests that DAAs might enhance patients’ immune response to HCC.

### 2.5. The Treatment with DAAs Enhanced Immune Responses by Decreasing the Frequency of PD-1-Expressing CD4+ and CD8+ T Cells

To confirm whether T cell responses to TAA-derived peptides were associated with phenotypic changes in T cells, T cell profiles were comparatively evaluated among the four groups (Figure 4A, Appendix A and Appendix A). The variation tendencies of the cell profiles in the four groups were different, especially between the increased and decreased group (Appendix A).

Interestingly, the frequency of PD-1+CD4+ and PD-1+CD8+ T cells significantly decreased after the treatment with DAAs only observed in the increased group (Figure 4A). In the increased group, the frequency of PD-1-expressing T cells decreased after the treatment with DAAs in the whole CD4+ or CD8+ T cell and each memory T cell fraction (Figure 4B). PD-1-expressing CD8+ memory T cells has been shown to be highly capable of producing IFN-γ in the ELISPOT assay [18]. Taken together, these results suggest that the treatment with DAAs decreased the frequency of PD-1 expressing T cells in patients to enhance immune responses to TAA-derived peptides.

### 2.6. The F Protein Increased the Frequency of PD-1-Expressing CD4+ or CD8+ T Cells

Our previous study identified genes whose expression is altered in peripheral blood cells, including lymphocytes, during treatment with DAAs for chronic HCV. An RNAseq analysis revealed that the mRNA expression level of the TbX21 gene increased after the treatment in many patients with SVR; however, a significant difference was not observed due to the insufficient number of cases analyzed (Appendix A). TbX21 has been shown to downregulate the expression of PD-1 in lymphocytes [19], while the expression of TbX21 was downregulated by the F protein produced by HCV [20,21]. F protein, a novel protein, is the product of an alternative reading frame (ARF) of the HCV polyprotein frame. In HCV genotype 1b, F protein produced by double frame-shift mechanism of the HCV structural protein core genomic region [20].

Based on these findings, we hypothesized that the elimination of HCV by DAAs may promote a decrease in or the disappearance of the F protein, which may, in turn, restore the expression of TbX21 in lymphocytes and downregulate that of the PD-1 molecule. To prove this hypothesis, we attempted to generate the F protein produced by HCV and analyze the phenotype of T cells after a mixed culture of the F protein and PBMCs.

We synthesized the genotype 1b HCV F protein component in Escherichia coli, consisting of the wild-type HCV core sequence with the first ribosomal frameshift at its codon 42(+1) and then second one at codon 144(−1) leading to creating a stop codon at 144 with a histidine tag (Figure 5A). We then confirmed this product by Western blotting using the direct detection of His-tagged proteins (Figure 5B) and the core protein (Figure 5C).

The synthetic F protein was approximately 17 kDa and was incubated with PBMCs collected from three cohorts: a healthy cohort (n = 22), HCV-infected cohort (n = 23), and SVR-obtained cohort (n = 9). The synthetic F protein significantly increased the frequency of PD-1-expressing CD4+ or CD8+ T cells in all three cohorts (Figure 6A,B).

We also compared the frequency of PD-1-expressing T cells in 8 patients for whom we could obtain sufficient PBMCs before and after the treatment with DAAs (Figure 6C). After the treatment, the frequency of PD-1-expressing T cells mostly decreased, whereas a significant increase was observed after the incubation with the F protein. Collectively, these results indicated that the F protein was positively associated with the frequency of PD-1-expressing CD4+ or CD8+T cells.

## 3. Discussion

To elucidate the influence of HCV eradication by DAAs for the immune response to HCC in HCV-infected patients and the mechanism underlying it, we first examined T cell responses to synthetic peptides derived from HCC-related TAAs. The number of patients with a positive immune response to each peptide was significantly higher after than before the treatment with DAAs (Figure 1A). There is no significant difference in immune responses to HCC-specific TAA-derived peptides between patients with and without a history of HCC and eradication of HCV. Furthermore, positive responses to the peptides GPC3_298_ and SCCA_112_ were newly induced after the treatment. These results suggest that treatment with DAAs enhances the immune response of T cells against HCC.

We next investigated the mechanisms by which T cell immune responses to HCC were enhanced. In previous studies, we demonstrated that the T cell profile in peripheral blood was associated with the immune strength of these cells against HCC-specific TAA-derived peptides [17,18,22]. To confirm whether different immune responses induced by DAAs are associated with immune phenotypic changes, we herein compared the expression of surface molecules on each immune cell before and after treatment. We focused on CD4+ and CD8+ T cells, which play important roles in altering immune responses. After the treatment with DAAs, the frequency of T cells expressing 6 out of the 9 surface molecules, which were PD-1, CTLA-4, CD80, OX40, CD25, and CXCR3, changed significantly, suggesting that the host T cell profiles were extensively altered by the treatment. The frequency of PD-1- or CTLA-4-expressing CD4+ or CD8+ T cells significantly decreased. PD-1 and CTLA-4 together with their ligands have been shown to play fundamental roles in the inhibition of T cell activation against tumors [23,24,25]. Furthermore, changes in T cell surface markers were more pronounced in patients with SVR than in those with non-SVR. These results indicate that the treatment with DAAs down-regulated the expression of these immunosuppressive molecules, particularly PD-1, in order to enhance specific immune responses to HCC.

To clarify the relationship between changes in both peptide-specific immune responses and the frequency of PD-1-expressing T cells before and after the treatment with DAAs, immune cell profiles were analyzed within four groups classified by changes in immune responses to TAA-derived peptides before and after the treatment with DAAs. In contrast to the decreased, mixed, and unchanged groups, the frequency of PD-1-expressing CD4+ and CD8+ T cells significantly decreased in the increased group only, in which the definite enhancement of immune responses was observed after treatment. Many studies have suggested that PD-1 suppresses immune responses by inhibiting T cell activation [24], and that the upregulated expression of PD-1 in CD4+ or CD8+ T cells is positively associated with T cell exhaustion and immune evasion [26,27]. Therefore, the significant decrease observed in the frequency of PD-1-expressing T cells in the increased group in the present study was consistent with the enhancement of TAA-derived peptide-specific immune responses. This result supports the treatment with DAAs enhancing the immune responses of patients to HCC by decreasing the frequency of PD-1-expressing CD4+ and CD8+ T cells.

The HCV-derived F protein has been reported to contribute to the persistence of HCV infection, which may cause hepatic damage and, ultimately, carcinogenesis [21,28,29] and through the PD-1/PD-L1 pathway to induce T cell dysfunction: accelerating cell apoptosis and impairing T cell proliferation [30]. To clarify whether the F protein influences the expression of PD-1, our synthesized F protein was incubated with PBMCs collected from three representative cohorts. After the incubation, the frequency of PD-1-expressing CD4+ or CD8+ T cells significantly increased in all three groups. This result suggests that the F protein plays a role in upregulating the expression of the PD-1 molecule to suppress the immune responses of patients.

Unexpectedly, in the present study, 2 out of the 3 non-SVR patients were categorized into the increased group based on their enhanced immune responses. When we examined the frequency of PD-1-expressing CD4+ T cells in these two patients after the treatment with DAAs, it was unchanged in one and decreased in the other patient. More importantly, the frequency of PD-1-expressing CD8+ T cells decreased in both of them. These results suggest that despite DAAs treatment did not eliminate HCV, it might downregulate the expression of PD-1 by decreasing or even eliminating the F protein. This may be one of the reasons for the enhanced immune response in non-SVR patients.

One limitation of the present study is that the HCV F-protein was not measured in patient sera before or after the treatment with DAAs. Further studies on the relationships between F protein levels in patient sera, the intensity of TAA-specific immune responses, and changes in T cell profiles before and after the treatment with DAAs are warranted to elucidate the mechanisms underlying the recovery of host immune responses to HCC by DAAs. Nevertheless, the present results indicate that the treatment with DAAs decreased the frequency of PD-1-expressing CD4+ and CD8+ T cells and improved immune responses to HCC-specific TAA-derived peptides, suggesting the involvement of the HCV-derived F protein.

## 4. Materials and Methods

### 4.1. Study Population

Patients diagnosed with chronic hepatitis C and receiving a combination of asunaprevir (ASV) (Bristol-Myers Squibb, New York, NY, USA) and daclatasvir (DCV) (Bristol-Myers Squibb) for 24 weeks were included in the present study. Patients with decompensated liver cirrhosis and infected with hepatitis B or human immunodeficiency virus were excluded. Patients received an oral dose of 100 mg ASV twice daily and 60 mg DCV once daily for 24 weeks. During this period, basic liver function indexes, such as ALT and the amount of HCV RNA, were monitored. Serum HCV RNA levels were measured using a real-time PCR method with the lower quantification limit of 1.2 log IU/mL (COBAS TaqMan HCV Test 2.0; Roche Diagnostics, Tokyo, Japan). A sustained virologic response (SVR) was defined as undetectable HCV RNA at 24 weeks post-treatment. The human leukocyte antigen (HLA) typing of patients was performed using peripheral blood mononuclear cells (PBMCs) and the PCR–reverse-sequence-specific oligonucleotide method. HLA-A24-positive patients were included in the present study. All patients provided written informed consent to participate, and the study protocol conformed to the ethical guidelines of the 1975 Declaration of Helsinki and was approved by the regional Ethics Committee (Medical Ethics Committee of Kanazawa University, No. 1639).

### 4.2. Preparation of Synthetic Peptides and PBMCs

HLA-A24-restricted peptides were synthesized using the amino acid sequences derived from 14 HCC-related TAAs as previously reported (Table 2) [17,31,32,33,34,35,36,37,38,39,40,41,42,43]. In addition, the HLA-A24-restricted peptide derived from cytomegalovirus pp65 (CMV pp65_328_) was synthesized [44]. Peripheral blood samples were obtained from 47 patients before treatment with DAAs and 24 weeks after the completion of treatment. PBMCs were isolated according to a previously described procedure [17], resuspended in Roswell Park Memorial Institute 1640 medium (RPMI-1640) containing 80% fetal bovine serum (FBS) and 10% dimethylsulfoxide (Sigma, St. Louis, MO, USA), and cryopreserved until used.

### 4.3. IFN-γ ELISPOT Assay

ELISPOT assays were performed as previously described [17]. Specific spots were calculated by subtracting average spots of control wells from the average spots of each peptide treated duplicate wells. Responses for peptides were considered positive if the number of specific spots was ≥10 and at least two times that of spots in the control group. The pattern of changes in the frequency of T cells specific for TAA-derived peptides after the treatment with DAAs was classified as following. A significant increase was defined as the number of specific spots was ≥10 after treatment and two-fold higher than that before treatment. In contrast, a significant decrease in the number of specific spots was defined as a pre-treatment spot count of 10 or more, and a post-treatment spot count of less than one-half of the pre-treatment spot count.

Based on the pattern of changes in the frequency of T cells specific for TAA-derived peptides after the treatment with DAAs, patients were classified into four groups. Patients who showed an increased frequency of T cells specific for at least one TAA-derived peptide and no decrease in the frequency of T cells specific for other peptides after treatment were classified into the increased group. Patients who showed a decrease in the frequency of T cells specific for at least one TAA-derived peptide and no increase in the frequency of T cells specific for other peptides were classified into the decreased group. Patients who showed a mixed frequency of increases and decreases in T cells specific for TAA-derived peptides were classified into the mixed group. Patients who did not show an increase or decrease in the frequency of T cells specific for any peptide were classified into the unchanged group.

### 4.4. Multicolor Fluorescence-Activated Cell Sorting Analysis

To examine differences in cell profiles before and after the treatment with DAAs, a flow cytometry analysis was performed. Isolated PBMCs, which were the same as the samples used for the ELIPOST assay, were stained by different antibodies and then examined using flow cytometry with the Becton Dickinson FACSAria II system. Based on previously reported data, a FACS analysis was performed using the following antibodies: anti-CD45RA, Foxp3, CD3, CD4, CD8, PD-1, CTLA-4, CD25, CCR6, CXCR3, CCR4, 4-1BB, OX40, and CD80 (BD Biosciences, Franklin Lakes, NJ, USA) [7,8,17,22]. The antibody CD45RA was used to divide CD4 or CD8 into naïve and memory T cells. Intracellular staining with anti-Foxp3 was conducted to characterize regulatory T cells (Tregs). In addition, the cell profiles of MDSCs were analyzed using anti-CD14, CD15, CD33, CD11b, HLA-DR, and PD-L1.

### 4.5. Expression and Purification of the HCV F Protein

The process of F protein expression was performed as previously described [45]. Briefly, the synthetic HCV F protein component consisted of the wild-type HCV core sequence and the HCV F protein, which included a ribosomal frameshift mutation located at codon 42 and stop codon at codon 144 of the core sequence. Overall, the polyprotein was partially shortened. Purified F PCR products were cloned into the cold shock expression system, the pCold I DNA vector (Takara, Japan) upstream of the six-His-tagged tail. The combined plasmid and empty vector as a negative control was transformed into BL21 cells and transformants were selected on an agar plate containing ampicillin that was shaken with 5 mL LB medium at 37 °C until OD_600_ reached 0.4–0.8. The culture was quickly cooled to 15 °C using ice water and left to stand for 30 min. IPTG was added at a final concentration of 0.2 mM and the culture was shaken at 15 °C for 24 h. Pelleted bacteria (5000 rpm, 10 min) were resuspended in xTractor^TM^ buffer Kit (Takara, Japan) containing 8M urea, disrupted by sonication, and incubated at 4 °C overnight. The lysate was then centrifuged at 8000 rpm at 4 °C for 1 h. The Capturem™ His-Tagged Purification Kit (Takara, Japan) was used to purify the F protein. The concentration of the purified F protein was 12 ug/uL. The recombinant F protein was verified by a Western blot analysis and the direct detection of His-tagged proteins using Nickel-NTA conjugates. The protein synthesized from the empty vector was a negative control and the 29 kDa protein PA tag (carboxy-terminal)-EGFP-6×His tag (Wako, Japan) was a positive control.

The purified F protein was incubated with PBMCs isolated from three cohorts: a healthy cohort (n = 22), HCV-infected cohort (n = 23), and SVR-obtained cohort (n = 9), at a final concentration of 12 µg/mL for 24 h, and the frequency of PD-1 expressed on CD4+ or CD8+ T cells was examined by FACS. Each incubation was performed in triplicate.

### 4.6. Western Blotting

Proteins were separated on a 15% SDS-PAGE gel (Wako, Japan) and then were transferred to a PVDF membrane (Millipore, Burlington, MA, USA). To detect His-tagged proteins, the membrane was blocked by complete immersion in 1X Detector Block Solution (HisDetector^TM^ Western Blot Kit, HRP Colorimetric) at room temperature for 1 h. After blocking, the HisDetector Nickel-AP, which was used to detect His-tag protein, was diluted to 1/1000 directly in blocking solution and then incubated at room temperature for one hour. After that, 1X PBST was used to wash the membrane 3 times. The membrane was then incubated in BCIP/NBT and allowed to develop for 5–15 min. The membrane was rinsed for 10–30 s in reagent quality water to stop the reaction. To detect the core protein, the membrane was blocked with 5% skim milk solution (Wako, Japan) for 1 h and then incubated with a Hepatitis C Virus Core Antigen Monoclonal Antibody (C7-50) (Thermo Fisher Scientific, Rockford, IL, USA) diluted 1:1000 overnight. The membrane was subsequently incubated for 1 h with anti-mouse IgG second antibody (Cell Signaling Technology, Danvers, MA, USA) diluted 1:1000. The enhanced chemiluminescent HRP substrate detection kit (Bio-Rad, Hercules, CA, USA) was used to visualize reactive protein bands.

### 4.7. Statistical Analysis

Data were expressed as the median and IQR. Statistical analyses were performed with GraphPad Prism 9.0.0 (GraphPad Software, San Diego, CA, USA). In the present study, the Student’s *t*-test and chi-squared test were used. The Student’s paired *t*-test was used to compare data between before and after the treatment with DAAs and with or without the synthetic F protein incubation. *p* < 0.05 was considered to be significant. Asterisks were used to indicate significance as follows: * *p*  <  0.05, ** *p*  <  0.01, *** *p*  <  0.001, and **** *p* < 0.0001.

## 5. Conclusions

In conclusion, the present results suggest that the treatment with DAAs enhances the immune responses of patients to HCC, and these changes contribute to the prevention of hepatocarcinogenesis after DAAs treatment.

## Figures and Tables

**Figure 1 ijms-23-11623-f001:**
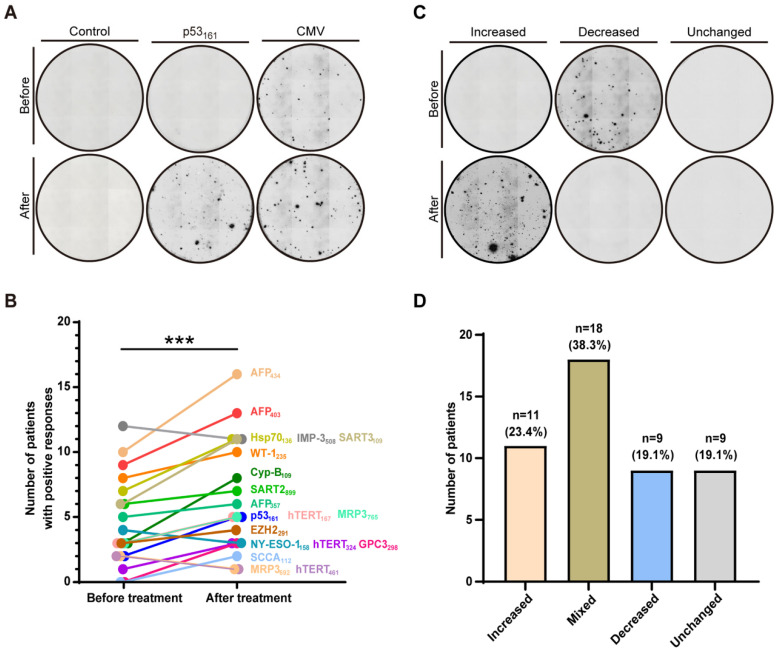
HCC-related TAA-derived peptide-specific immune responses. (**A**) Representative images of IFN-γ ELISPOT assay testing PBMCs stimulating and unstimulating with TAAs (figure shows p53_161_) as well as before and after antiviral therapy. The IFN-γ ELISPOT assay was performed to examine immune responses to 19 TAA-derived and CMV pp65 control peptides in 47 patients before and after the treatment with DAAs. (**B**) Dot plot showing the number of patients with a positive response to each TAA-derived peptide before and after the treatment with DAAs, detected by the IFN-γ ELISPOT assay. The paired *t*-test was used to calculate *p* values. (**C**) Representative images of the IFN-γ ELISPOT assay showing significant increase, significant decrease as well as unchanged response. (**D**) The number of patients showing four different patterns of changes in the frequency of T cells specific for TAA-derived peptides after the treatment with DAAs. Patients were categorized into the following four groups, as described in the Materials and Methods: increased group (n = 11), mixed group (n = 18), decreased group (n = 9), and unchanged group (n = 9). *** *p* < 0.001.

**Figure 2 ijms-23-11623-f002:**
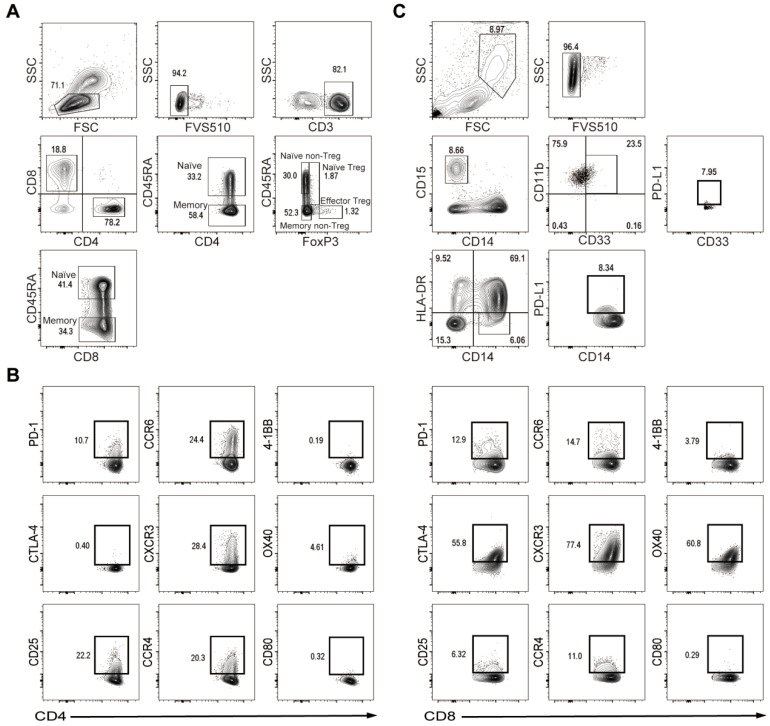
Peripheral blood T cell and MDSC profiles. (**A**) Flow cytometry analysis of CD4+ and CD8+ T cells. Subpopulations of CD8+ T cells were defined by the expression of CD45RA: naïve CD8+ T cells (CD8+CD45RA+) and memory CD8+ T cells (CD8+CD45RA-), and subpopulations of CD4+ T cells by the expression of CD45RA: naïve CD4+ T cells (CD4+CD45RA+) and memory CD4+ T cells (CD4+CD45RA-). After the further division of CD4+ T cells by marker Foxp3, naïve non-Treg CD4+ T cells were defined as CD4+CD45RA+Foxp3-, naïve Tregs as CD4+CD45RA+Foxp3+, memory non-Treg CD4+ T cells as CD4+CD45RA-Foxp3-, and effector Tregs as CD4+CD45RA-Foxp3++. (**B**) The expression of the following molecules was separately analyzed in each subpopulation of CD4+ or CD8+ T cells: PD-1, CTLA-4, CD25, CCR6, CXCR3, CCR4, 4-1BB, OX40, and CD80. (**C**) Subpopulations of MDSCs were also analyzed by flow cytometry. G-MDSC were characterized by CD14-CD15+CD33+CD11b+, and M-MDSC by CD14+HLA-DR-. PD-L1 expression levels were measured in each subpopulation. Abbreviations: FSC, forward scatter; SSC, side scatter.

**Figure 3 ijms-23-11623-f003:**
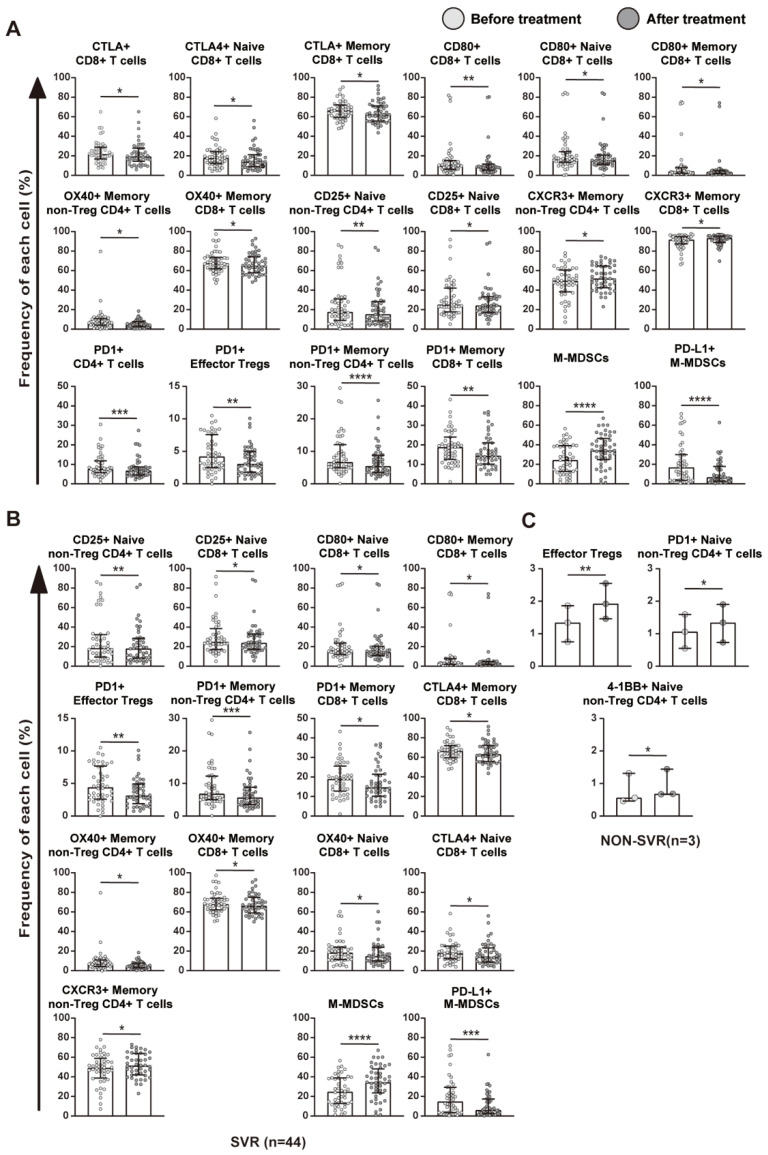
Changes in T cell and MDSC profiles before and after the treatment with DAAs. (**A**) Comparative analysis of the frequencies of each T cell and MDSC subpopulation before and after the treatment with DAAs by flow cytometry in all patients (n = 47). (**B**) Comparison of the frequencies of each T cell and MDSC subpopulation before and after the treatment with DAAs in the SVR group (n = 44). (**C**) Comparison of the frequencies of each T cell and MDSC subpopulation before and after the treatment with DAAs in the non-SVR group (n = 3). The figure only shows results with a significant difference. Box plots depict the median value with IQR. The paired *t*-test was used to calculate *p* values. * *p* < 0.05, ** *p* < 0.01, *** *p* < 0.001, **** *p* < 0.0001.

**Figure 4 ijms-23-11623-f004:**
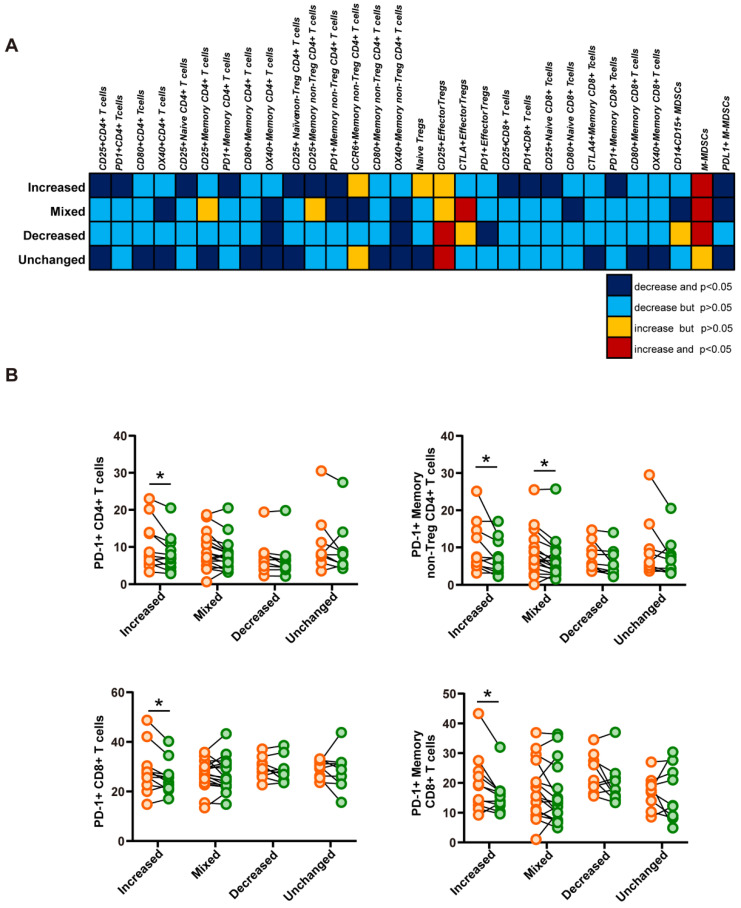
Relationship between TAA-derived peptide-specific immune responses and the frequency of immune cells with each immune phenotype. (**A**) Representative phenotypic variation map before and after treatment within the four groups classified based on the results of the ELISPOT assay. In comparisons with before the treatment with DAAs, changes in the frequency of T cells expressing each molecule after treatment are indicated separately by four different colors. (**B**) Dot plot displaying the frequency of PD-1-expressing T cells before (orange circle) and after (green circle) the treatment with DAAs within the four groups. The paired t-test was used to calculate p values. * *p* < 0.05.

**Figure 5 ijms-23-11623-f005:**
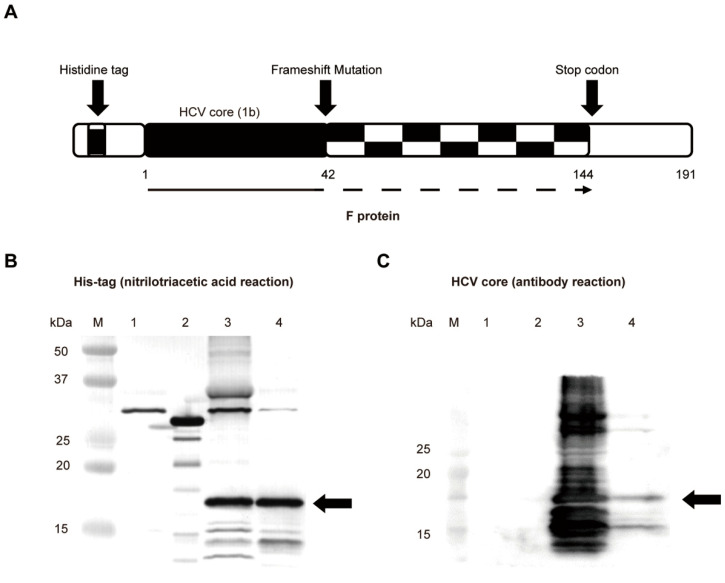
Confirmation of the in vitro synthesized F protein by a Western blot analysis. (**A**) The structure of the HCV F protein expressed in *Escherichia coli*. It consisted of the wild-type HCV core sequence and the HCV F protein that includes a frameshift mutation at 42 and the introduction of a histidine tag. This results in the induction of a stop codon at 144, and the overall polyprotein was partially shortened. (**B**) Western blotting images of the purified HCV F protein detected by the His tag. The image shows His-tagged bands by a nickel-nitrilotriacetic acid reaction. (**C**) Western blotting images of purified proteins detected by the HCV core antibody with the epitope located within codons 1–42. The black arrow shows the synthesized HCV-F protein band. M, molecular marker; 1, negative control: empty vector; 2, positive control for the His tag: PA tag-EGFP-6XHis tag; 3, non-purified F protein; 4, His-tagged purified F protein.

**Figure 6 ijms-23-11623-f006:**
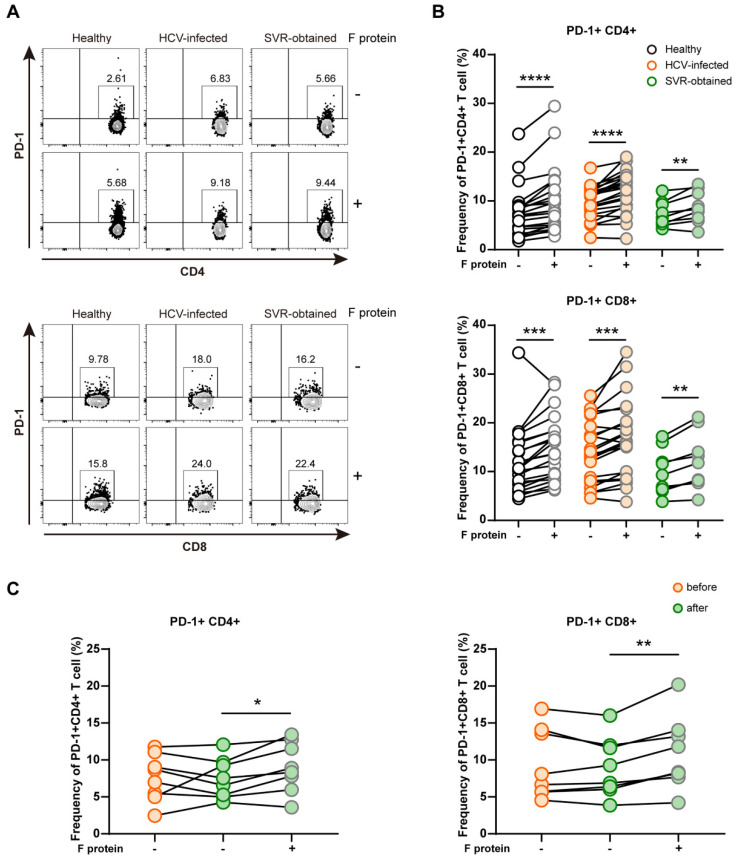
Changes in the frequency of PD-1-expressing CD4+ and CD8+ T cells in PBMCs before and after the stimulation with the HCV F protein. (**A**) Representative flow cytometry plots of PD-1+CD4+ T cells and PD-1+CD8+ T cells following an ex vivo stimulation with or without the F protein. Each incubation was performed in triplicate. The analysis was performed using the PBMCs of a healthy cohort (n = 22), HCV-infected cohort (n = 23), and SVR-obtained cohort (n = 9). (**B**) Dot plot displaying the frequency of PD-1-expressing CD4+ and CD8+ T cells in PBMCs with or without the F protein in three cohorts. (**C**) Dot plot displaying the frequency of PD-1-expressing CD4+ or CD8+ T cells in the PBMCs of 8 paired chronic hepatitis C patients before treatment with DAAs, after treatment without the F protein stimulation, and after treatment with the F protein stimulation. Statistical analyses were performed using the paired *t*-test. * *p* < 0.05, ** *p* < 0.01, *** *p* < 0.001, **** *p* < 0.0001.

**Table 1 ijms-23-11623-t001:** Clinical characteristics of HCV-infected patients before treatment with DAAs.

	Participants (N = 47)		
Characteristic	DAAs for 24 Weeks		
	SVR (n = 44)	Non-SVR (n = 3)	*p* Value *
Age, years median ± SD	65 ± 9.66	65 ± 6.25	NS
Sex, M/F	15/29	0/3	NS
BMI (kg/m^2^), median ± SD	22.48 ± 3.48	23.15 ± 0.51	NS
HCV RNA, median ± SD	6.0 ± 0.64	6.2 ± 0.15	NS
IL28B, N (%)			NS
Major	22(50)	2(67)	
hetero	21(48)	1(33)	
minor	1(2)	0(0)	
With cirrhosis, N (%)	19(43)	2(67)	NS
With HCC, N (%)	6(14)	0(0)	NS
L31 (NS5A) (+), N (%)	1(2)	0(0)	NS
Y93 (NS5A) (+), N (%)	0(0)	2(67)	0.003
ALT(IU/L)	48.9 ± 34.58	48.67 ± 24.01	NS
AST(IU/L)	53 ± 34.60	38.7 ± 15.04	NS
HCV treatment naive, N (%)	14(32)	1(33)	NS

Abbreviations: SVR, sustained virologic response; IL28B, interleukin-28B gene; NS5A, nonstructural protein 5A; ALT, alanine transaminase; AST, aspartate transaminase; AFP, alpha-fetoprotein; HbA1c, glycated hemoglobin; (+), mutation; NS, not significant. * The Student’s *t*-test, chi-squared test (Fisher’s exact test).

**Table 2 ijms-23-11623-t002:** The list of TAA-derived peptides analyzed.

Peptide No.	Peptide Name	Source	Reference	Amino Acid Sequence
1	Cyp-B_109_	Cyp-B	[31]	KFHRVIKDF
2	SART2_899_	SART2	[32]	SYTRLFLIL
3	SART3_109_	SART3	[33]	VYDYNCHVDL
4	p53_161_	p53	[34]	AIYKQSQHM
5	MRP3_765_	MRP3	[35]	VYSDADIFL
6	MRP3_692_	MRP3	[35]	AYVPQQAWI
7	AFP_403_	AFP	[36]	KYIQESQAL
8	AFP_434_	AFP	[36]	AYTKKAPQL
9	AFP_357_	AFP	[36]	EYSRRHPQL
10	hTERT_167_	hTERT	[37]	AYQVCGPPL
11	hTERT_461_	hTERT	[37]	VYGFVRACL
12	hTERT_324_	hTERT	[37]	VYAETKHFL
13	WT-1_235_	WT-1	[38]	CYTWNQMNL
14	EZH2_291_	EZH2	[39]	KYDCFLHPF
15	GPC3_298_	GPC3	[40]	EYILSLEEL
16	NY-ESO-1_158_	NY-ESO-1	[41]	LLMWITQCF
17	SCCA_112_	SCCA	[42]	TYLFLQEYL
18	IMP-3_508_	IMP-3	[17]	KTVNELQNL
19	Hsp70_136_	Hsp70	[43]	GYPVTNAVI
20	CMV pp65_328_	CMV pp65	[44]	QYDPVAALF

Abbreviations: Cyp-B, cyclophilin B; SART, squamous cell carcinoma; MRP3, multiple drug resistance protein 3; AFP, alpha-fetoprotein; hTERT, human telomerase reverse transcriptase; WT1, Wilms tumor 1; EZH2, enhancer of zeste homolog 2; GPC3: glypican-3; NY-ESO-1: New York esophageal squamous cell carcinoma-1; SCCA: squamous cell carcinoma antigen; IMP-3: insulin-like growth factor II mRNA-binding protein 3; Hsp70, 70-kDa heat shock protein; CMV, cytomegalovirus.

## Data Availability

Data will be made available upon request.

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
