# Peer review of "Alterations in Hepatocellular Carcinoma-Specific Immune Responses Following Hepatitis C Virus Elimination by Direct-Acting Antivirals"

_ijms, 2022, doi:10.3390/ijms231911623_

Round 1

Reviewer 1 Report

HCV CD4+ and CD8+ specific T cells responses are functionally impaired during chronic hepatitis C infection. DAAs therapies eradicate HCV infection in more than 95% of treated patients. However, the impact of HCV elimination on immune responses remain controversial. In this study, Li et al. investigated the immune responses to HCC influenced by DAAs in patients with chronic hepatitis C. Although the results were clinically interesting, several points need be addressed.

1.     Regarding the study design, the inclusion and exclusion criteria of patients should be described. For example, whether HBV and HCV coinfection was excluded.

2.     Six patients with a history of HCC were included in this study. The authors should explain the reason to include patients with HCC.

3.     Is immune responses to HCC specific TAA-derived peptides different in patients with and withut history of HCC?

4.     Is immune responses to HCC specific TAA-derived peptides different in patients with and withut cirrhosis?

Reviewer 2 Report

Comments to authors

Shihui Li and the colleagues submitted the manuscript entitled ‘Alterations in hepatocellular carcinoma-specific immune responses following hepatitis C virus elimination by direct-acting antivirals’ This manuscript is very interesting. But, there are some issues to be addressed.

1. Authors analyzed the 47 chronic HCV-infected patients who received ASV and DCV. Among these patients, there were 6 patients who had a history of HCC prior to treatment with DAAs and 3 patients who did not achieve SVR. Why are these patients included in the present study? Please authors discuss the influence between the history of HCC and the present results and between the non-SVR and the present results.

2. There were 19 patients with cirrhosis. Are HCC-specific immune response differed between the patients with cirrhosis and those with non-cirrhosis?

3. Is the frequency of the development of HCC differed during follow-up after SVR between the four groups (the increased, mixed, decreased, and unchanged group) ?
